# Combined Metabolome and Transcriptome Analysis Highlights the Host’s Influence on *Cistanche deserticola* Metabolite Accumulation

**DOI:** 10.3390/ijms24097968

**Published:** 2023-04-27

**Authors:** Ru Feng, Hongshuang Wei, Rong Xu, Sai Liu, Jianhe Wei, Kun Guo, Haili Qiao, Changqing Xu

**Affiliations:** Institute of Medicinal Plant Development, Chinese Academy of Medicinal Science and Peking Union Medicinal College, Beijing 100193, China

**Keywords:** *Cistanche deserticola*, *Haloxylon ammodendron*, root-parasitic plant, medicinal plant cultivation, phenylethanoid glycoside, host–parasite interaction

## Abstract

The medicinal plant *Cistanche deserticola* Ma (Orobanchaceae) is a holoparasitic angiosperm that takes life-essential materials from *Haloxylon ammodendron* (C. A. Mey.) Bunge (Amaranthaceae) roots. Although many experiments have been conducted to improve the quality of *C. deserticola*, little attention has been paid to the host’s influence on metabolite accumulation. In this study, transcriptomic and metabolomic analyses were performed to unveil the host’s role in *C. deserticola*’s metabolite accumulation, especially of phenylethanoid glycosides (PhGs). The results indicate that parasitism by *C. deserticola* causes significant changes in *H. ammodendron* roots in relation to metabolites and genes linked to phenylalanine metabolism, tryptophan metabolism and phenylpropanoid biosynthesis pathways, which provide precursors for PhGs. Correlation analysis of genes and metabolites further confirms that *C. deserticola*’s parasitism affects PhG biosynthesis in *H. ammodendron* roots. Then we found specific upregulation of glycosyltransferases in haustoria which connect the parasites and hosts. It was shown that *C. deserticola* absorbs PhG precursors from the host and that glycosylation takes place in the haustorium. We mainly discuss how the host resists *C. deserticola* parasitism and how this medicinal parasite exploits its unfavorable position and takes advantage of host-derived metabolites. Our study highlights that the status of the host plant affects not only the production but also the quality of Cistanches Herba, which provides a practical direction for medicinal plant cultivation.

## 1. Introduction

Cistanches Herba is a traditional Chinese medicine. In the Chinese Pharmacopoeia, Cistanches Herba is described as ‘the dried fleshy stem of *Cistanche deserticola* Ma or *Cistanche mongolica* Beck (previously named *Cistanche tubulosa*) (*Orobanchaceae*)’. Chinese medicine practitioners prefer using materials derived from *C. deserticola* because it has been distributed in a broader area including all of northwestern China throughout history and has, thus, been used as a tonic for a longer time. As a holoparasite, this plant loses photosynthetic capacity and takes nutrients as well as water from its host. In previous studies by our group, *C. deserticola* mostly parasitized *Haloxylon ammodendron* (C. A. Mey.) Bunge (Amaranthaceae) (Figure 1). We also put *C. deserticola* seeds around the roots of *Haloxylon persicum* Bunge ex Boiss. et Buhse (Amaranthaceae) plants, and they successfully developed haustoria. In recent years, *Atriplex canescens* (Pursh) Nutt. (Amaranthaceae) has been reported as a new host for *C. deserticola* [1]. Many studies have investigated the factors that influence Cistanches Herba quality, and the results demonstrate a complex mechanism relying on multi-factor cooperation. For example, soil affects the quality of Cistanches Herba in at least two aspects: First, the composition and function of soil microbiomes contribute to quality variation of *C. deserticola* in different ecotypes [2]. Secondly, the wound stress caused by soil friction triggers the accumulation of phenylethanoid glycosides [3]. *C. deserticola* samples collected in spring and autumn also differ in their phenylethanoid glycoside contents [4].

However, little attention has been paid to the host plant’s effect on the quality of *C. deserticola*, which cannot survive if it is separated from the host. Studies in other medicinal phytoparasites demonstrate that host plants significantly influence their quality. For example, *Thesium chinense* Turcz. (*Santalaceae*) plants have higher concentrations of total amino acids when parasitizing *Triticum aestivum* L. (*Poaceae*) than when parasitizing other hosts [5]. Another example of the host’s effect on quality is *Scurrula ferruginea* (Jack) Danser (*Loranthaceae*): *S. ferruginea* stems have different anti-inflammatory properties when parasitizing on three different host plants [6].

Phenylethanoid glycosides (PhGs) are the major active components in Cistanches Herba. PhGs have been reported to have strong antioxidant potential [7] and hypoglycemic/hypolipidemic [8] and anti-tumor [9] effects. In terms of phytochemical taxonomy, the co-occurrence of PhGs with iridoids is a characteristic of *Lamiales* species [10]. As iridoids in Orobanchaceae parasites are observed to be sequestered from the host [11], we wondered whether PhGs would exhibit the same behavior.

The aglycons of PhGs are derived from the tyrosine metabolism pathway. Tyrosol, the aglycon of salidroside and osmanthuside H, is derived from tyrosine by two-step or three-step enzymatic reactions [12]. Most PhGs have a feruloyl or caffeoyl moiety in the glucose backbone, and their antioxidant potential is related to these structures [13]. Caffeic acid and ferulic acid are derived from the phenylpropanoid pathway. Compared with tyrosine metabolism, this pathway has been elucidated more clearly due to the fact that it is the upstream pathway of several resistant metabolites in plants. 

In this study, we compared *C. deserticola* fleshy stems, their hosts’ roots, and healthy *H. ammodendron* roots at the metabolomic and transcriptomic levels. By doing so we aimed to (1) explore the defensive changes in *H. ammodendron* roots caused by *C. deserticola* parasitism, (2) mark the candidate strategies used by *C. deserticola* to reduce host-derived damage, and (3) determine how this kind of host–parasite interaction influences *C. deserticola* metabolite accumulation, especially PhGs. Our research highlights the host’s influence on the quality of Cistanches Herba and proposes a direction for improvement in the field of medicinal plant cultivation.

## 2. Results

### 2.1. Metabolic Profiling of C. deserticola Fleshy Stems and H. ammodendron Roots

We collected *C. deserticola* fleshy stems, their hosts’ roots, and healthy *H. ammodendron* roots (Table 1) from the Bencaocongrong Planting Base, Ningxia, China. All the *H. ammodendron* plants at this base were planned to be the hosts of *C. deserticola*, and *C. deserticola* seeds had been sown around their roots. In the fall of 2019, we dug holes around the trees to find out whether they had been parasitized and chose nine parasitized trees and nine healthy trees to collect plant materials. In the summer of 2020, we followed the same procedure.

Based on the LC-MS/MS results, the metabolites in *C. deserticola* and its host plant *H. ammodendron* could be classified into 13 kinds, among which phenolic acids and lipids constitute a large proportion (Figure 2). Hierarchical cluster analysis (HCA) of the metabolites showed good intragroup repeatability as well as obvious intergroup difference (Appendix A). The results of HCA also revealed that the CS cluster is a short distance from the HcS cluster, indicating a similarity between the parasite and its hosts. However, the CS cluster was far from the HS cluster, which had no physical connection with *C. deserticola*. Thus, the observed similarity was a consequence of parasitism. The principal component analysis (PCA) results showed significantly different accumulation and expression patterns of metabolites from *C. deserticola* fleshy stems and from the host and healthy *H. ammodendron* roots. Notably, HcF-vs.-HF and HcS-vs.-HS could be significantly separated by PCA, emphasizing the changes caused by parasites (Appendix A).

### 2.2. Differently Accumulated Metabolites in C. deserticola Fleshy Stems and H. ammodendron Roots

To screen the differentially accumulated metabolites in *C. deserticola* fleshy stems and *H. ammodendron* roots (CF-vs.-HcF, HcF-vs.-HF, CS-vs.-HcS and HcS-vs.-HS), an orthogonal partial least squares discrimination analysis (OPLS-DA) was performed (Appendix A). Based on the OPLS-DA results, metabolites were selected based on the thresholds of VIP ≥ 1 and ABS (fold change) ≥ 2, and 264, 159, 397, and 234 DAMs were found in CF-vs.-HcF, HcF-vs-HF, CS-vs.-HcS and HcS-vs.-HS, respectively. The DAMs were then annotated and assigned to KEGG pathways. The KEGG pathways for which CF-vs.-HcF DAMs were enriched significantly were tyrosine metabolism and tryptophan metabolism (Figure 3A). The significant enrichment pathways of CS-vs.-HcS were tyrosine metabolism and sulfur metabolism (Figure 3B). These results indicate significant differences in metabolites linked to the tyrosine metabolism pathway, from which phenylethanoids are derived, for *C. deserticola* fleshy stems and roots of its host. Interestingly, most DAMs from the tyrosine metabolism pathway had relatively low concentrations in *C. deserticola* except for rosmarinic acid and salidroside. The latter is a typical PhG derived from tyrosol. The KEGG pathways for which HcF-vs.-HF DAMs were significantly enriched were phenylalanine metabolism and tryptophan metabolism (Appendix A). Five DAMs linked to phenylalanine metabolism: succinic acid, tyrosine, salicylic acid, benzoic acid, and phenylalanine. All of them were upregulated, and tyrosine was upregulated the most (8.60-fold). DAMs in the tryptophan pathway were upregulated except for N-acetyl-5-hydroxytryptamine, which was downregulated 0.00046-fold. The significant enrichment pathways of HcS-vs.-HS were pyruvate metabolism, glycolysis/gluconeogenesis, fructose/mannose metabolism, and propanoate metabolism (Appendix A).

### 2.3. Phenylethanoid Glycosides and Their Precursors in C. deserticola Fleshy Stems and H. ammodendron Roots

New PhGs continue to be discovered, but not all of them have been included in online databases such as KEGG, and their complete biosynthetic pathways are unclear. Thus, we identified PhGs and candidate precursors according to the previous literature and compared their relative concentrations in *C. deserticola* fleshy stems and *H. ammodendron* roots.

In this study, we identified 19 PhGs in total (Appendix A). Nine of them took hydroxytyrosol as their aglycon. Among them, cistanoside H had the simplest structure, and only appeared in *H. ammodendron* roots in the summer (Figure 4A). Echinacoside appeared in all groups of samples, but its concentration in healthy *H. ammodendron* roots (HF and HS) was relatively low. 2’-Acetylacteoside was detected in *C. deserticola* fleshy stems (CF) and their hosts’ roots (HcF) in the fall. The other six PhGs were only detected in *C. deserticola* fleshy stems. Two PhGs took tyrosol as their aglycon and six PhGs took homovanillyl alcohol. PhGs that took tyrosol as their aglycon (salidroside, osmanthuside H) had simpler structures and appeared in *C. deserticola* fleshy stems (CS), their hosts’ roots (HcS) and healthy roots (HS) (Figure 4B). Cistanoside E bonded no caffeoyl or feruloyl groups, and it could be detected in CF, HcF, HF, CS, and HcS. On the other hand, the other five PhGs that took homovanillyl alcohol as their aglycon showed more complexity in their structures and were limited in distribution. Two PhGs contained other aglycons. Isocrenatoside was only detected in CF samples and isomartynoside in both CF and HcF samples. In a word, most PhGs were distributed only in *C. deserticola* fleshy stems, and those that appeared in both hosts and healthy *H. ammodendron* roots often had simpler structures, except for echinacoside and cistanoside A.

The precursors for PhGs could be classified as precursors for (1) the phenylethanoid aglycon and (2) the phenylpropanoid group modifying the glucose backbone. The former is linked to the tyrosine metabolism pathway and showed significantly different metabolite accumulation patterns between the parasite and host. The latter is produced from the phenylpropanoid biosynthesis pathway. In *C. deserticola* fleshy stems and *H. ammodendron* roots, no hydroxytyrosol or homovanillyl alcohol was detected. Tyrosol was distributed in all samples but maintained relatively lower levels in CF and CS (Figure 5). As polyphenol oxidase could also take glycosides as its substrate, PhGs such as echinacoside would be derived from simpler structures such as salidroside and cistanoside H. Dopamine, another candidate precursor for phenylethanoid aglycon, was only detected in *H. ammodendron* roots. The phenylpropanoid groups are often derived from caffeic acid or ferulic acid. Both phenylpropanoids showed relatively lower concentrations in CF and CS.

### 2.4. Transcriptomic Changes Caused by Parasitism of C. deserticola in H. ammodendron Roots

Besides the redistribution of metabolites, the parasitism of *C. deserticola* also caused significant transcriptomic changes in *H. ammodendron* roots. A total of 3291 differentially expressed genes were screened in the HcF-vs.-HF group, with 1280 upregulated and 2011 downregulated (Appendix A). A total of 3992 DEGs were screened in the HcS-vs.-HS group, with 2116 upregulated and 1876 downregulated (Appendix A). KEGG pathway enrichment analysis revealed that DEGs in the HcF-vs.-HF group were significantly enriched in metabolic pathways, phenylpropanoid biosynthesis, starch/sucrose metabolism, flavonoid biosynthesis, and fatty acid elongation (Appendix A). The KEGG pathways for which HcS-vs.-HS DEGs were significantly enriched included terpenoid backbone biosynthesis, linoleic acid metabolism, phenylpropanoid biosynthesis, biosynthesis of secondary metabolites, alpha-linolenic acid metabolism, the MAPK signaling pathway, plant–pathogen interaction, isoquinoline alkaloid biosynthesis, isoflavonoid biosynthesis, and cyanoamino acid metabolism (Appendix A). In both seasons, the expression levels of genes linked to phenylpropanoid biosynthesis were significantly changed in parasitized *H. ammodendron* roots compared with healthy ones. This indicates that the phytoparasite functions as a biotic stress. Similar to viruses and insects, invasion by the phytoparasite stimulates the host’s defensive response, and the phenylpropanoid biosynthesis pathway is one of the key pathways functioning in plant defense.

### 2.5. Genes Specifically Upregulated in Haustoria

The haustorium connects the phytoparasite and its host. Many reactions take place here and they have a great impact on the parasitic system’s homeostasis. To identify genes specifically upregulated in haustoria, we compared the expression data of haustoria with *C. deserticola* fleshy stems and *H. ammodendron* roots. DEGs were screened in the XS-vs.-CS and XS-vs.-HcS groups and enriched in KEGG pathways. In the XS-vs.-CS group, 25,842 DEGs were screened and 23,940 were up-regulated in the XS samples (Appendix A). The DEGs were significantly enriched in plant hormone signal transduction, the MAPK signaling pathway of the plant, benzoxazinoid biosynthesis, terpenoid backbone biosynthesis, alpha-linolenic acid metabolism, and arachidonic acid metabolism (Appendix A). In the XS-vs.-HcS group, 28,174 DEGs were identified and 27,490 were upregulated in the XS samples (Appendix A). The DEGs were significantly enriched in plant hormone signal transduction and the proteasome (Appendix A). In conclusion, the haustoria exhibited certain gene expression patterns that were significantly different from both the parasite and the host, and this specific expression pattern may take part in the parasitic system’s hormone signal transduction.

Though a large number of DEGs were screened, few genes were specifically upregulated in the XS samples. The Venn plot shows that 2880 unigenes were specifically upregulated in the haustoria (Figure 6A). These unigenes were translated into amino acid sequences and uploaded to the Mercator 4.0 online annotation tool. The annotation results reveal that these specifically upregulated genes are mainly linked to bins such as enzyme classification, RNA biosynthesis, solute transport, protein homeostasis, and phytohormone action (Figure 6B). Within the bin of “enzyme classification”, most sequences were annotated as transferases, mainly transferring phosphorus-containing groups and glycosyl groups (Figure 6C and Appendix A). Phosphate is a decisive element in the parasitism of *Orobanchaceae* plants, and the relatively high glycosylation activity echoed the metabolomics results. Transporting solutes is another key function of the haustoria, and the annotation results showed that carrier-mediated transport was the main mode (Appendix A). The phytohormone action in haustoria involved common hormones such as cytokine and auxin, as well as those especially important to phytoparasites such as strigolactone (Appendix A). In a word, the haustorium functions as a center for metabolism, transportation, and signal transduction in the parasitic system, and it influences both the parasite and the host.

### 2.6. Integrated Analysis of Metabolites and Transcripts Linked to Phenylethanoid Glycoside Biosynthesis in H. ammodendron Roots

To investigate the differences in phenylethanoid glycoside biosynthesis caused by the parasitism of *C. deserticola*, transcriptomic and metabolomics data of *H. ammodendron* roots were analyzed together (Figure 7). Nine PhGs were detected in *H. ammodendron* roots: echinacoside, isomartynoside, osmanthuside H, salidroside, 2′-acetylacteoside, cistanoside A, cistanoside E, cistanoside H and epimeridinoside A. L-phenylalanine, L-tyrosine, tyrosol, L-dopa, ferulic acid, dopamine, and caffeic acid were found to be involved in phenylethanoid glycoside biosynthesis. In the HcF-vs.-HF group, dopamine, L-phenylalanine, L-tyrosine, tyrosol, echinacoside, isomartynoside, 2′-acetylacteoside, and cistanoside A were screened as DAMs, and DAMs in the HcS-vs.-HS group included tyrosol, cistanoside E, cistanoside H, osmanthuside H, and epimeridinoside A. In the HcF-vs.-HF group, 22 DEGs were identified as candidates for phenylethanoid glycoside biosynthesis (Appendix A): *C4H* (2DEGs), *4CL* (1DEG), *C3*′*H* (1DEG), *CCR* (7DEGs), and *UGT* (11DEGs). In the HcS-vs.-HS group, 54 DEGs were found: *4CL* (3DEGs), *C3*′*H* (1DEG), *CCR* (12DEGs), *HCT*(3DEGs), *TyDC*(2DEGs), *PAO*(3DEGs), *PPO*(5DEGs), and *UGT* (24DEGs).

To further understand how DEGs and DAMs are involved in phenylethanoid glycoside biosynthesis, Pearson’s correlation analysis was performed (Appendix A). In the HcF-vs.-HF group, the accumulation levels of echinacoside, 2′-acetylacteoside, and astanoside A were positively correlated with two *UGT* genes. Another phenylethanoid glycoside, isomartynoside, was also correlated with two *UGT*s. Meanwhile, the expression levels of *C3*′*H*, *C4H*, and *UGT* were positively correlated with tyrosol and dopamine. In the HcS-vs.-HS group, the accumulation levels of osmanthuside H, astanoside E, and astanoside H were positively correlated with two *4CL*, three *CCR*, one *PPO*, and three *UGT* genes. Six DEGs were correlated with tyrosol, including one *PPO*, one *CCR*, and four *UGT*s.

### 2.7. Identification and Phylogenetic Analysis of UGTs in C. deserticola and H. ammodendron

As shown by the metabolomic and transcriptomic results, glycosylation plays an important role in the communication between parasite and host. We identified UGTs in *C. deserticola* and *H. ammodendron* and performed a phylogenetic analysis to find out their possible functions. First, we annotated 124 unigenes as UGTs in *C. deserticola* and 258 unigenes in *H. ammodendron*. Then, unigenes encoding >300 amino acid proteins were selected and a Blastp analysis was performed. This yielded 74 and 134 UGTs in *C. deserticola* and *H. ammodendron*, respectively (Appendix A). In *C. deserticola*, the UGTs were separated into 18 families. Among them, the UGT74 family was the largest with 12 members, followed by UGT85 with 8 members. In *H. ammodendron*, there were 19 UGT families; the UGT85 family, which was the largest, had 15 members.

Then we aligned the UGTs in *C. deserticola* and *H. ammodendron* with representative sequences from *Arabidopsis thaliana* to construct a phylogenetic tree. These UGTs were phylogenetically separated into 13 conserved groups (A–M) identified in Arabidopsis and 1 novel group (P) identified in maize (Figure 8). The *C. deserticola* UGTs were incorporated into these groups except for Group N. Among them, eight UGT85 family members contributed to the construction of Group G, which was the largest *C. deserticola* UGT group. No Group J members were found in *H. ammodendron* UGTs. Group A was the largest *H. ammodendron* group, including 10 UGT91 family members and 5 UGT79 family members, followed by Group G with 13 UGT85 family members.

### 2.8. UGT Candidates Involved in Phenylethanoid Glycoside Biosynthesis

Combining the results of Blastp and phylogenetic analysis, we determined that 8 unigenes in *C. deserticola* and 12 unigenes in *H. ammodendron* showed 70–83% identity to characterized UGT85As and were phylogenetically related to the representative AtUGT85A1. Though most UGTs exhibited a broad substrate selectivity, UGT85A1 was found to prefer tyrosol as its most suitable substrate for salidroside production. The significantly positive correlation between Cluster-57997.0 and isomartynoside also confirmed the role of UGT85A subfamily members in PhG biosynthesis (Appendix A). The other three PhGs differentially accumulated in HcF-vs.-HF (echinacoside, 2′-acetylacteoside, and cistanoside A) were positively correlated with Cluster-27477.1, which was 63% identical to *A. thaliana* UGT73B5 and phylogenetically related to the representative AtUGT73B3.

### 2.9. Quantitative RT-PCR Validation for C. deserticola and H. ammodendron Genes

Ten genes were selected to validate the accuracy of transcriptomic data using qRT-PCR analysis, including five *C. deserticola*, and five *H. ammodendron* genes. In *C. deserticola*, *CYP98A*, *CYP96A*, *CYP35A*, *ABC-T* (ABC transporter), and *BG* (1,3-beta-glucosidase) were selected. In *H. ammodendron*, *C3H*, *HCT*, *4CL*, *TyDC*, and *PPO* were selected. The results were consistent with the transcriptome data (Figure 9A,B).

## 3. Discussion

*Orobanchaceae* root-parasitic plants take life-essential materials from their hosts, which causes significant damages to the hosts. Sunflowers (*Helianthus annuus* L.) infected by the holo-parasitic broomrape (*Orobanche cumana*) have larger and thinner leaves because of carbohydrate depletion [14]. Reduced height and weight have been observed in tobacco (*Nicotiana* spp.) parasitized by *O. cernua* [15]. Though retaining certain biosynthetic capacities, hemiparasitic plants do not stop stealing from autotrophic plants. The parasitism of witchweed (*Striga* spp.) has a significant impact on the head index, leaf biomass, leaf index, root biomass, root index, plant biomass, and root: shoot ratio of sorghum (*Sorghum bicolor* L. Moench) [16]. *Rhinanthus minor* has a direct negative influence on host productivity and competitive ability, which helps reconstruct plant communities in grassland [17]. To avoid or reduce losses, different strategies are used based on the developmental stages of the parasite. Before the haustorium is established, the main task is to prevent seed germination and haustorium initiation. Suppressing seed germination stimulants (in most cases strigolactones) biosynthesis and enhancing cell wall lignification are characteristic of pre-haustorial resistance [18,19]. If that fails and the haustorium attaches to the host root, the focus shifts to deconstructing the vascular connection and producing a toxic environment [20]. In the case of *O. crenata*, increased phenolic compounds in host plants are related to limited parasitic damage, which can be induced by salicylic acid and indole acetic acid [21]. A similar phenomenon is observed in sunflowers resistant to *O. cumana*, which is associated with the growth restriction of the parasite [22].

A more interesting phenomenon is the parasite’s response. Lignin monomers in plants can be classified into four types: H-unit, C-unit, S-unit, and G-unit. The S-unit lignifies the cell wall of sclerenchyma fibers and sclereids, whereas the G-unit contributes to the construction of tracheary elements [23]. In research comparing *Phtheirospermum japonicum* and *Striga hermonthica*, the haustorium of *S. hermonthica* could be initiated by both the S and G-unit, whereas that of *P. japonicum* was only induced by the S-unit [24]. That is, *P. japonicum* senses the root of its gymnosperm host which has no tracheary elements and initiates the haustorium. Thus, the consequence of cell wall lignification is contrary to the initial goal of host resistance. Lignin monomers are not the only host-derived resistant chemicals utilized by parasites. In *Castilleja levisecta*, iridoid glycoside concentrations differ depending on host species and distance. The host-derived diversity leads to different survival rates of *Euphydryas editha* feeding on *C. levisecta*, which is partially related to the iridoid glycosides sequestration [25]. The situation is similar for *Orobanche ramose*, which sequestrates pyridine alkaloids from its *Nicotiana* hosts, and *Pedicularis semibarbata*, which obtains quinolizidine alkaloids from *Lupinus fulcratus* [26,27]. In a word, parasitic plants make full use of materials from the hosts, even including those initially designed as weapons.

In our study, the metabolic changes brought about by *C. deserticola* parasitism were linked to phenylalanine metabolism, tryptophan metabolism, pyruvate metabolism, glycolysis/gluconeogenesis, fructose/mannose metabolism, and propanoate metabolism. The fructose and mannose metabolism pathway is critical to the success of parasitism. In the case of *Orobanche aegyptiaca*, the host-induced suppression of the mannose 6-phosphate reductase gene is concomitant with significant mannitol decrease and increased tubercle mortality [28]. In plant–pathogen interaction, the pathogen secretes mannitol as a buffer against oxidative stress, and the host plant activates mannitol dehydrogenase to counter it [29]. The relatively low mannitol level in parasitized *H. ammodendron* root is a consequence of this counteraction. However, mannitol also acts as an osmoprotectant. The most evident difference between parasitic plants and fungi is that the former takes nutrients from the host mainly by vascular bundles, which may be directed by osmotic potential. Thus, the gap in mannitol levels between *C. deserticola* and *H. ammodendron* contributes to the reversed source–sink relationship and the efficient uptake of nutrients. This is another good example of the parasite’s utilization of the host’s resistance. In addition, PhG biosynthesis can be enhanced by mannitol [30]. The similar accumulation patterns of PhGs and mannitol in *H. ammodendron* roots confirmed this.

At the transcriptomic level, the parasitism of *C. deserticola* caused significant changes linked to the phenylpropanoid pathway in *H. ammodendron* root. The phenylpropanoid pathway is the upstream pathway where lignin, lignan, coumarin, and flavonoid are derived [31]. These metabolites help to build physical and chemical barriers. By utilizing them, plants under abiotic/biotic stress can protect themselves. Activation of the phenylpropanoid pathway can be regarded as *H. ammodendron*’s defense response to the parasite. The mitogen-activated protein kinase (MAPK) signaling pathway is also triggered by parasitism. The MAPK signaling pathway is involved in multiple defense responses, including plant stress/defense hormone biosynthesis/signaling, reactive oxygen species (ROS) generation, stomatal closure, defense gene activation, phytoalexin biosynthesis, cell wall strengthening, and hypersensitive response (HR) cell death [32]. In *H. ammodendron* root parasitized by *C. deserticola*, most upstream genes in the MAPK signaling pathway were significantly upregulated, whereas downstream ones were inhibited. This indicates that the defense response is suppressed by parasite-derived effectors. The downregulation of genes linked to plant–pathogen interaction also demonstrated this phenomenon.

*Orobanchaceae* parasites sequester secondary metabolites from hosts, including polyacetylenes, alkaloids, and iridoid glycosides. These metabolites help in resisting herbivores and other stresses [11]. The situation is more complicated for PhGs. While holoparasites retain the capacity of PhG synthesis, their PhG accumulation is influenced by the host. In the case of *O. laxissima*, the acteoside content is significantly influenced by the host’s acteoside content [33]. In our study on *Cistanche mongolica*, the host species had no significant influence on PhG contents. However, the height and weight of the parasites were significantly affected by the host species, and PhG contents decreased as the parasite grew. No PhGs have been tested in *Tamarix*, but precursors such as caffeic, coumaric, and ferulic acids exist in *Tamarix* plants [34]. In this research, we found that the genes linked to PhG biosynthesis were expressed in *H. ammodendron* and were significantly upregulated by the parasite. The existence of PhGs in healthy *H. ammodendron* roots also confirmed its capacity to synthesize these metabolites as well as their precursors. The PhGs cistanoside H and osmanthuside H showed relatively low concentrations in the host roots compared with the healthy roots. This may also be the result of the parasite’s depletion of precursors. Further proof of the host’s precursor feeding was significantly lower concentrations of caffeic acid, tyrosol, and dopamine in *C. deserticola*.

Then we questioned why *C. deserticola* changed the structure of host-derived phenols. Phenylpropanoids such as caffeic acid, ferulic acid, and coniferyl-alcohol can damage various cellular components, and their toxicity decreases when bonded to glucose [35]. Consuming precursors also inhibits the host’s cell wall lignification for more efficient intrusion [23]. Although tyrosol toxicity in plants is not mentioned in the literature, research on rotifer (*Philodina acuticornis*) reveals that tyrosol is more toxic to salidroside [36]. Dopamine is an important hormone in animals, and it also plays a vital role in the plant kingdom. Endogenous dopamine in plants is involved in abiotic/biotic stresses such as drought, salinity, nutritional deficiency, and pathogens [37]. By reconstructing host-derived tyrosol and dopamine, the parasite inhibits the host’s resistance and creates a more comfortable environment for its invasion. The large number of UGT 85A members expressed in *C. deserticola* supported this point, as UGT85A family members catalyze the glycosylation of tyrosol [38]. In *H. ammodendron*, the PhG content was significantly related to one UGT85A and one UGT73B gene expression level. UGT73B is also linked to the glycosylation of tyrosol, though the product can be icariside. This explains the UGT expression in the haustorium, where the parasite builds a vascular connection with the host. Therefore, by improving glycosylation, the parasite detoxifies compounds from the host and protects itself. The PhGs, which are considered as bioactive compounds in *C. deserticola*, play a vital role in this self-protection process.

## 4. Materials and Methods

### 4.1. Plant Materials

The plant materials were collected from the Bencaocongrong Planting Base, Ningxia, China. The *C. deserticola* fleshy stems, *H. ammodendron* roots, and haustoria were harvested from the plant using sharp scissors and cleaned using ultra-pure water. Materials from 3 plants were mixed as a biological replicate and each group had 3 replicates. All the samples were rapidly frozen by liquid nitrogen and transported to the Institute for Medicinal Plant Development, Peking Union Medicinal College, Beijing, China, in boxes filled with dry ice. Then the samples were stored at −80 °C for metabolomic, transcriptomic, and gene expression analyses.

### 4.2. Analysis of Metabolites in C. deserticola Fleshy Stems and H. ammodendron Roots

For this analysis, 6 groups of samples with 3 biological replicates per group were vacuum-dried by vacuum freeze-dryer (Scientz-100F) and mashed with a mixer mill (MM 400, Retsch, Shanghai, China) with a zirconia bead for 1.5 min at 30 Hz. Then, 100 mg of the lyophilized powder was dissolved in 1.2 mL 70% methanol solution, vortexed 6 times for 30 s at intervals of 30 min, and kept at 4 °C overnight. After that, the solution was centrifuged at 12,000 rpm for 10 min and the extract was filtrated through a microporous membrane (SCAA-104, 0.22 μm pore size; ANPEL, Shanghai, China, http://www.anpel.com.cn/ (accessed on 15 March 2021)) for further analysis. To analyze the metabolites, an UPLC-ESI-MS/MS system (UPLC: Shimadzu Nexera X2, www.shimadzu.com.cn/ (accessed on 15 March 2021); MS: Applied Biosystems Q TRAP 4500, www.appliedbiosystems.com.cn/ (accessed on 15 March 2021)) was used [39,40]

Based on the mass spectrometry data, metabolites were identified using the Metware Database (MWDB, Wuhan, China) and quantified according to peak intensity. The quantification data of metabolites were normalized by unit variance scaling and used for PCA and HCA. OPLS-DA was performed after log2 transformation and mean centering. These analyses were carried out using R 3.6.3 software (http://www.r-project.org/; accessed on 20 November 2021) [41].

### 4.3. Screening of Differentially Accumulated Metabolites

To determine the metabolomic differences between *C. deserticola* and its host, differentially accumulated metabolites in the CF-vs.-HcF and CS-vs.-HcS groups were screened using the mass spectrometry data. VIP values were extracted from OPLS-DA results and metabolites with VIP ≥ 1 and absolute log2 (fold change) ≥ 1 were selected as differentially accumulated metabolites [42].

The DAMs were annotated using the KEGG Compound database (http://www.kegg.jp/kegg/compound/ (accessed on 25 March 2021)) and mapped to the KEGG Pathway database (http://www.kegg.jp/kegg/pathway.html (accessed on 25 March 2021)) [43]. Then a KEGG pathway enrichment analysis was performed, and the significance was determined by hypergeometric test *p*-values.

To investigate the changes caused by *C. deserticola* parasitism, the same procedure was followed for the HcF-vs.-HF and HcS-vs.-HS groups.

### 4.4. RNA Extraction and Illumina Sequencing

Total RNA was extracted from the plant tissues using a quick RNA isolation kit (DP452, Tiangen, Beijing, China). To ensure the RNA samples were integrated and DNA-free, agarose gel electrophoresis was performed. RNA purity was then determined by a nanophotometer. Following that, a Qubit 2.0 Fluorometer and an Agilent 2100 BioAnalyzer were used to accurately measure RNA concentration and integrity, respectively. The qualified samples were processed with oligo (dT) beads to enrich the mRNA, which was broken into fragments and used as templates for the cDNA library. To qualify the cDNA library, the fluorometer was used for primary quantification and the bioanalyzer was then used to insert text size. A q-PCR analysis was then performed to accurately quantify the library. The qualified library was sequenced using the Illumina HiSeq 4000 platform.

### 4.5. RNA-Seq Analysis of C. deserticola and H. ammodendron

Clean reads were obtained by eliminating low-quality reads and assembled using Trinity 2.6.6 software [44]. The transcripts were assembled and then clustered into unigenes. To explore the genes related to haustorium functions, we put unigenes from the CS, XS, HcS, and HS groups into one library and determined the gene expression levels in the whole parasitic system. The unigenes were annotated using the KEGG, NR, SwissProt, Trembl, GO, and KOG databases. Gene expression levels were calculated in terms of reads per kb per million reads (RPKM).

The differentially expressed genes were screened based on the thresholds of FDR < 0.5 and absolute log2FC > 1 using DESeq2 1.22.2 software [45]. Then a KEGG pathway enrichment analysis was performed, and the significance was determined by the hypergeometric test’s *p*-values.

### 4.6. Correlation Analysis of Inigenes and Metabolites Linked to PhG Biosynthesis

DEGs and DAMs linked to PhG biosynthesis were selected to perform Pearson’s correlation tests using SPSS 26.0 software. Correlations with Pearson’s correlation coefficient (PCC) ≥ 0.7 and *p* ≤ 0.05 were uploaded to Cytoscape 3.9.1 software to draw connection networks between DEGs and DAMs [46].

### 4.7. Identification and Phylogenetic Analysis of UGTs

The candidate UGTs in *C. deserticola* and *H. ammodendron* were identified following a procedure described by Cheng et al. [47]. First, a hidden Markov model (HMM) was retrieved from the Pfam database (http://pfam.sanger.ac.uk (accessed on 1 November 2022)) and the UGTs were identified using HMMER 3.0. Then, the identified unigenes were processed by the Simple Modular Architecture Research Tool (SMART) (http://smart.embl-heidelberg.de (accessed on 1 November 2022)) and ORF Finder (http://bioinf.ibun.unal.edu.co/servicios/sms/orf_find.html (accessed on 1 November 2022)) for additional validation. Following that, genes encoding protein sequences < 300 amino acids long were removed. The UGTs were processed by MEGA 10.2 software for further analysis [48]. Multiple sequence alignment was conducted using the ClustalW program [49]. The maximum likelihood method was used to generate a phylogenetic tree, with 1000 replicates used for bootstrap testing to validate the tree [50]. ITOL (Interactive Tree Of Life, https://itol.embl.de (accessed on 1 December 2022)) was used to modify the tree [51].

### 4.8. Quantitative RT-PCR Validation of Differential Gene Expression

Ten unigenes were selected to validate the accuracy of the transcriptomic data using qRT-PCR analysis. Reverse transcription was performed using a MonScript™ RTIII All-in-One Mix kit with dsDNAse (MR05101; Monad, Wuhan, China). The primers for the 10 unigenes were designed using Primer Premier 6.0, and 2 internal controls (Appendix A) were prepared using sequences reported in the literature (Appendix A) [52,53]. qRT-PCR analysis was performed using the Applied Biosystems 7500 platform (Thermo Fisher, Waltham, MA, USA) and a QuantiNova SYBR Green PCR Kit (208054, Qiagen, Hilden, Germany). Three biological replicates were assessed in each group and three technical replicates were included in each qRT-PCR reaction. The 2−ΔΔCt method was used to calculate the relative expression levels [54].

## 5. Conclusions

In this study, host metabolites and genes changed by *C. deserticola* parasitism were determined. These metabolites and genes were linked to phenylalanine metabolism, tryptophan metabolism, and phenylpropanoid biosynthesis pathways, which provide precursors for PhGs. *C. deserticola* makes use of host-derived components to produce PhGs. We highlight the host’s influence on the quality of Cistanches Herba and recommend that *C. deserticola* growers select suitable host plants.

## Figures and Tables

**Figure 1 ijms-24-07968-f001:**
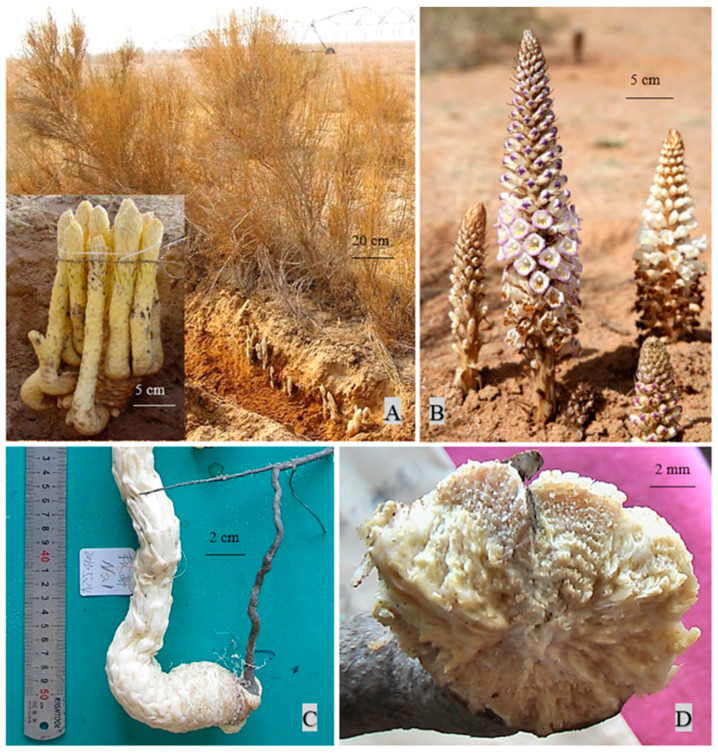
*Cistanche deserticola* Ma and its host plant. (**A**) *Cistanche deserticola* Ma fleshy stems collected for medicinal use and its host *Haloxylon ammodendron* (C. A. Mey.) Bunge. (**B**) *C. deserticola* inflorescence. (**C**) *C. deserticola* joins the host root. (**D**) Junction of *C. deserticola* with the host root (including haustorium).

**Figure 2 ijms-24-07968-f002:**
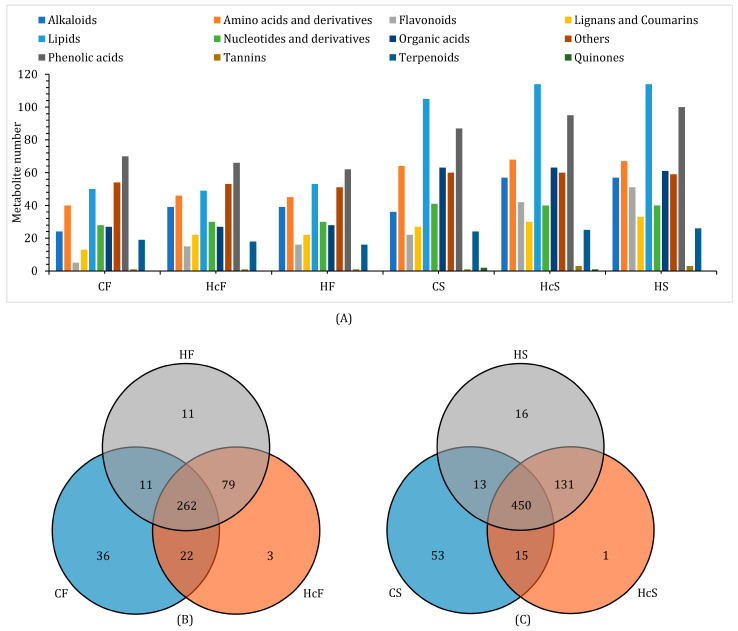
Summary of metabolites detected by using LC-MS/MS methods. (**A**) Classification of metabolites detected in *C. deserticola* fleshy stems, the hosts’ roots, and healthy *H. ammodendron* roots. (**B**) Venn plot revealing the relationship of metabolites in CF, HcF, and HF samples. The figures in circles represent the numbers of metabolites. (**C**) Venn plot revealing the relationship of metabolites in CS, HcS, and HS samples. The figures in circles represent the numbers of metabolites.

**Figure 3 ijms-24-07968-f003:**
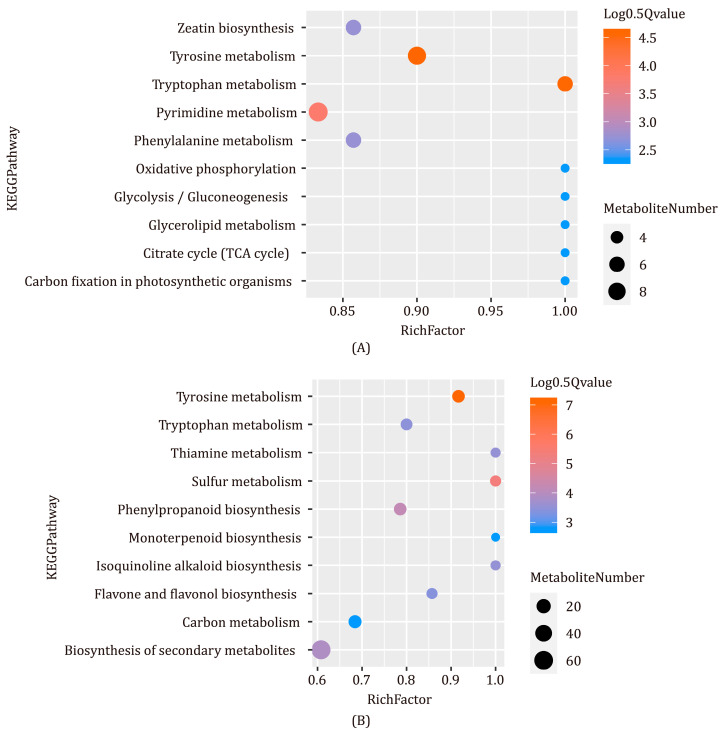
Bubble plots showing KEGG enrichment analysis results using DAMs in *C. deserticola* fleshy stems and the host *H. ammodendron* roots. The bubble’s color indicates the q value, and its size represents the number of DAMs enriched. (**A**) The bubble plot of the top 10 KEGG pathways to which CF-vs.-HcF DAMs were enriched. (**B**) The bubble plot of the top 10 KEGG pathways to which CS-vs.-HcS DAMs were enriched.

**Figure 4 ijms-24-07968-f004:**
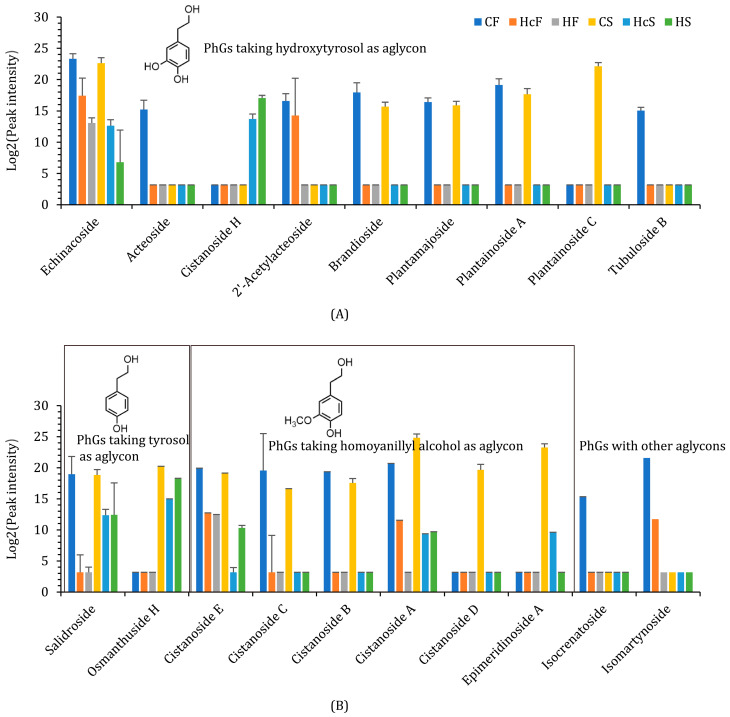
Relative contents of PhGs in *C. deserticola* fleshy stems, the host, and healthy *H. ammodendron* roots. The relative contents of PhGs were determined using peak intensity (mean ± SE, *n* = 3). (**A**) PhGs taking hydroxytyrosol as their aglycon. (**B**) PhGs taking other phenylethanoids as their aglycon.

**Figure 5 ijms-24-07968-f005:**
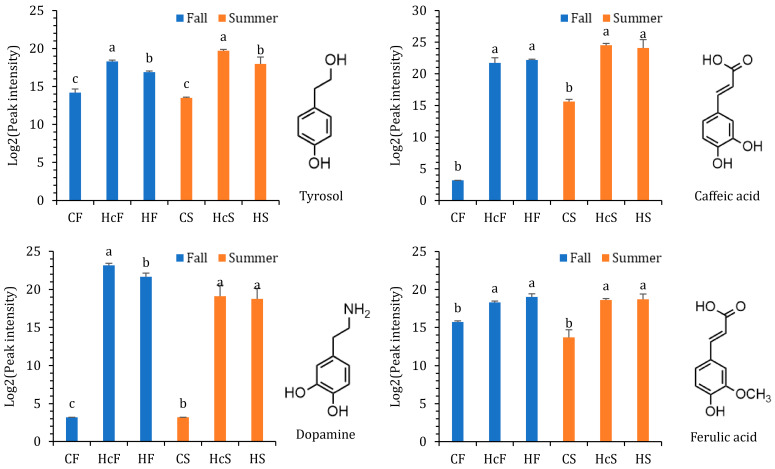
PhG precursors and their relative contents in *C. deserticola* fleshy stems, the host, and healthy *H. ammodendron* roots. The relative contents of PhG precursors were determined using peak intensity. Samples from different seasons were analyzed separately. Bars labeled with different letters are significantly different (mean ± SE, *n* = 3, Tukey’s HSD, *p* < 0.05).

**Figure 6 ijms-24-07968-f006:**
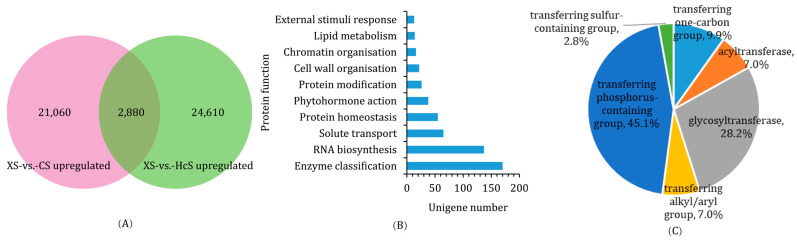
The screening and annotation of genes specifically upregulated in haustoria. (**A**) Venn plot showing the number of genes upregulated in XS samples compared with CS and HcS, respectively. The intersection indicates the number of genes specifically upregulated in haustoria. (**B**) The bar plot shows the top 10 Mercator classifications of genes specifically upregulated in haustoria. (**C**) The pie plot shows the classifications of genes annotated as ‘transferase’.

**Figure 7 ijms-24-07968-f007:**
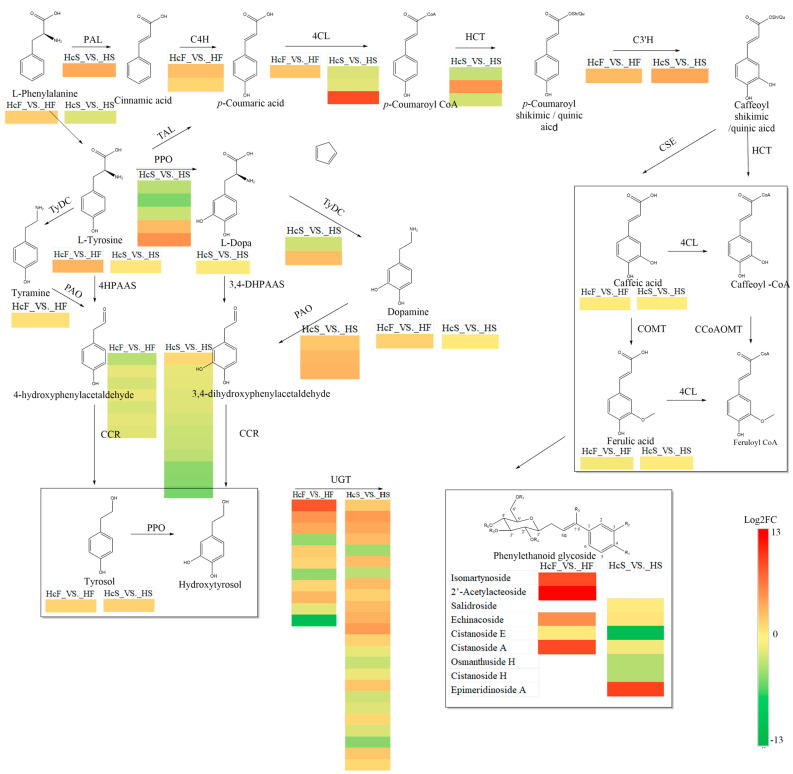
Phenylethanoid glycoside biosynthesis pathway in parasitized and healthy *H. ammodendron* roots. Grids with colors from green to red represent log2 fold changes in DEGs and metabolites (HcF-vs.-HF, HcS-vs.-HS). 4HPAAS, 4-hydroxyphenylacetaldehyde synthase; 4HPAR, 4-hydroxyphenylacetaldehyde reductase; TyDC, tyrosine decarboxylase; PPO, polyphenol oxidase; PAO, primary amine oxidase; CCR, cinnamyl-CoA reductase; PAL, phenylalanine ammonia-lyase; C4H, cinnamate 4-hydroxylase; 4CL, 4-coumarate: CoA ligase; HCT, p-hydroxycinnamoyl-CoA:qui-nate/shikimate p-hydroxycinnamoyltransferase; C3H, p-coumarate 3-hydroxylase; CSE, caffeoyl shikimate esterase; COMT, caffeic acid O-methyltransferase; CCoAOMT, caffeoyl-CoA O-methyltransferase.

**Figure 8 ijms-24-07968-f008:**
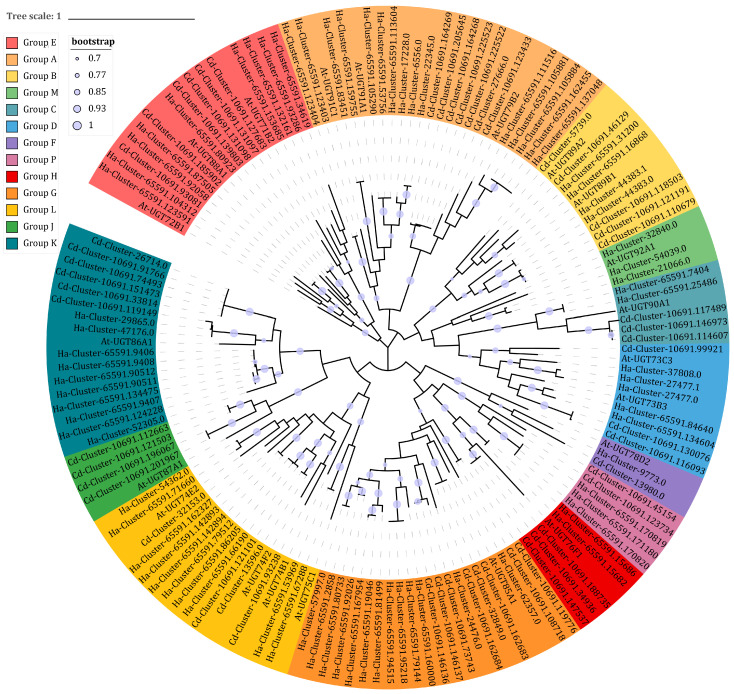
Phylogenetic tree of *C. deserticola* and *H. ammodendron* candidate UGTs using *A. thaliana* UGTs as representative sequences. Cd, *C. deserticola*; Ha, *H. ammodendron*; At, *A. thaliana*.

**Figure 9 ijms-24-07968-f009:**
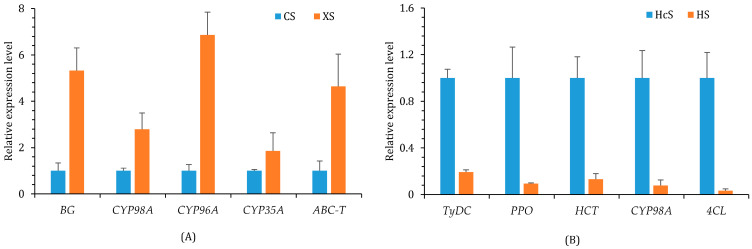
qRT-PCR validation for *C. deserticola* and *H. ammodendron* genes. (**A**) Relative expression levels of *CYP98A*, *CYP96A*, *CYP35A*, *ABC-T*, and *BG* genes in *C. deserticola.* (**B**) Relative expression levels of *C3H*, *HCT*, *4CL*, *TyDC*, and *PPO* genes in *H. ammodendron*.

**Table 1 ijms-24-07968-t001:** Information about plant materials.

	Sample ID	Species	Harvesting Time	Tissue
CF	CF1	*C. deserticola*	November 2019	Fleshy stem, 5 cm from haustorium
	CF2			
	CF3			
HcF	HcF1	*H. ammodendron*		Root intruded by *C. deserticola*, 10 cm from haustorium
	HcF2			
	HcF3			
HF	HF1			Root of healthy plant
	HF2			
	HF3			
CS	CS1	*C. deserticola*	August 2020	Fleshy stem, 5 cm from haustorium
	CS2			
	CS3			
HcS	HcS1	*H. ammodendron*		Root intruded by *C. deserticola*, 10 cm from haustorium
	HcS2			
	HcS3			
HS	HS1			Root of healthy plant
	HS2			
	HS3			
XS	XS1	*C. deserticola* and *H. ammodendron*		Haustorium, the connecting place of *C. deserticola* and *H. ammodendron*, with a thickness of 0.5 cm
	XS2			
	XS3			

## Data Availability

All data in this study is available from the corresponding author upon reasonable request.

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
