# Peer review of "Combined Metabolome and Transcriptome Analysis Highlights the Host’s Influence on Cistanche deserticola Metabolite Accumulation"

_ijms, 2023, doi:10.3390/ijms24097968_

Round 1

Reviewer 2 Report

Paper is related to the relationship of metabolites production and gene expression in Cistanche

The paper shows the transcriptome and metabolome and trying to do a correlationship between them

The study seems to be complete and show interesting conclusions

Author Response

Thank you for your suggestions.

We added conclusions in the end of the manuscript.

Reviewer 3 Report

Dear Authors,

Your manuscript entitled "Combined analysis of metabolome and transcriptome highlighted the host’s influence on Cistanche deserticola metabolites accumulation" is very interesting indeed. However, it suffers from one major problem - your English language must be improved significantly. Otherwise, it is really difficult to understand. Therefore, I must recommend a major revision and would like to review again a revised version with appropriately polished English both in terms of grammar and style.

A minor remark concerns the host specificity of Cistanche deserticola. I want to know whether this parasitic plant is strictly specific to the host Haloxylon ammodendron or can also infest other potential host species. This is important to be addressed in the Introduction section.  

Round 2

Reviewer 3 Report

Dear Authors,

Your manuscript was significantly improved. However, I will request some further changes, more specifically:

- Table 1 and the respective information in the text should be moved to Results section, it is not appropriate for the Introduction. For example the paragraph, starting on line 71 should start with: "In this study,  C. deserticola fleshy stems, their hosts’ roots and healthy H. ammodendron roots were compared by metabolomic and transcriptomic analyses...." Then move your Table with some more information in either Results or Material section.

- overall, your Introduction section is kind of short and disproportionate compared to the rest of the manuscript. I will appreciate a paragraph, or two, for example containing information on how host plant species affect specialized metabolites accumulation in other parasitic plants. 

- the start of your Discussion: "Orobanchaceae root-parasitic plants are crafty thieves. They sense signals from the hosts, germinate, intrude into the hosts’ roots, and develop vascular connections. Though all of these efforts their only goal is to steal life-essential materials which other plants spend their whole lives collecting. Such an unequal trade causes significant damages to the hosts." must be re-written. This is not literature or poetry, this is a scientific paper and you must adhere to certain standards.

- you are not consistent in citations: "the glycosylation of tyrosol[36]" is wrong, it must be "the glycosylation of tyrosol [36]", e.g., add intervals before the citation wherever needed.

- some minor stylistic and grammar polishing is required. 
